# The Potential of Amaranth as a Basic Raw Material for the Production of Pasta for a Vegan Diet

**Ilze Beitane *** **and Alla Marisheva**

Department of Nutrition, Faculty of Food Technology, Latvia University of Life Sciences and Technologies, 3004 Jelgava, Latvia
* Correspondence: ilze.beitane@llu.lv

**Featured Application: By improving the technology of amaranth pasta developed to obtain instant pasta, it would be possible to develop disaster food with high nutritional value.**

**Abstract:** The growing public interest in vegan products due to their association with health creates a need for new nutritious products that could cover nutritional deficiencies in the vegan diet. This study aimed to evaluate the potential of amaranth pasta and amaranth pasta blends with buckwheat or oat flour for a vegan diet to provide the limiting nutrients. Different types of pasta were developed in the current study—pasta from amaranth flour, and pasta samples from amaranth flour partly substituted with buckwheat or oat flour. The nutritional and energetic value, sugars, fatty acids, amino acids, and iron content were determined for all samples. All developed pasta samples can be evaluated as products with increased nutritional value, which provide more than 15% protein of the total energy with a sufficient content of lysine, methionine, and threonine; the iron content in the samples covered over 30% of the daily reference intake for iron; and the content of unsaturated fatty acids was above 70% of the total fat content. More nutritious pasta for a vegan diet can be obtained by substituting amaranth flour with buckwheat or oat flour. By improving the technology of amaranth pasta developed to obtain instant pasta, it would be possible to develop disaster food with high nutritional value.

**Keywords:** amaranth; buckwheat; oat; extrusion; energetic value; amino acids; fatty acids; iron; healthy nutrition

## 1. Introduction

Nowadays, more and more people adhere to a vegan lifestyle due to its association with health. A vegan diet is characterised by a low content of saturated fatty acids and a high content of polyunsaturated fatty acids and dietary fibre, which reduces the risk of mortality from ischemic heart disease [1]. Ischemic heart disease is known to be the leading cause of death worldwide regardless of income [2]. The results of a cohort study showed that the amount of daily energy consumed by vegans was 8127 kJ, of which carbohydrates provided 54.0%, total sugar 23.7%, protein 13.1%, fat 30.5%, and saturated fatty acids 6.9% [3]. Vegans had the lowest mean BMI at all age groups in both genders, 22.34 kg/m$^2$ in men and 21.75 kg/m$^2$ in women, compared to meat eaters, fish eaters, and vegetarians [4]. The mentioned study results, as well as the European Commission priority for 2019–2024: A European Green Deal [5], serve as a guarantee for the popularity of veganism. In response to growing demand, the number of vegan products on the market is increasing as well. A previous study of the supply of vegan products in Latvia showed the prevalence of soya (*Glycine max*) as a principal raw material for vegan food products [6]. Due to its versatility and use in both the human diet and forage, soybean is the most cultivated oil crop in the world. A growing demand suggests a need to expand the offer of vegan products on the market by finding alternative raw materials with the equivalent

nutrient content. The recent increase in interest for cultivating pseudo-cereals such as quinoa, buckwheat, and amaranth is justified by such grains' nutritional characteristics [7]. Despite the growing popularity in the research of these crops, their commercialisation remains quite undeveloped [8].

According to Market Research Report (2021), with the increasing consumer demand for organic food and focusing on a healthy lifestyle with an emphasis on plant-based products, the amaranth product sector is expected to experience a robust growth in North America, a moderate growth in Europe, and a significant growth in the Asia Pacific region [9]. Amaranth is cultivated in the countries of Asia, Africa, and America, and since the 1990s, in many European countries [10], including Latvia. Amaranth can be used in nutrition in different forms: whole seeds, flour, bran, and oil. Amaranth seeds have a nutty flavour and they can be prepared through boiling, roasting, crushing, or grinding. Amaranth seeds are a source of a rare type of starch (48–69%), high-quality fat (5.6–10.9%), and nutritionally valuable proteins (13.1–21.0%) [11,12]; they are suitable for a gluten-free diet. Amaranth seeds are distinguished from the common cereal grains by their high lysine content, which ranges from 4.9 to 6.1 g per 100 g of protein and which is limited in other cultures [13]. High amounts of fats, carbohydrates, fibre, sugars, and vitamins A, B, C, and E also make amaranth a good cereal grain alternative [14]. Amaranth surpasses the common cereals in potassium (3498 mg kg$^{-1}$), phosphorous (4360 mg kg$^{-1}$), and magnesium (2550 mg kg$^{-1}$) [15]; 100 g of seeds contain 173 mg of calcium, 35 mg of iron, and 3 mg of zinc, and such components as rutin, ferulic acid, and nicotiflorin make amaranth a potential health-promoting product [16,17]. The health benefits of amaranth are reported in the scientific literature—for instance, the decrease of plasma cholesterol level, stimulation of the immune system, anti-tumour activity, reduction of blood glucose level, positive effect on anaemia and hypertension [18,19]. Moreover, antioxidant and anti-allergic properties are also among amaranth's health-promoting qualities, from which patients with chronic diseases, allergies to common cereals, and celiac disease could benefit [18].

Amaranth flour is most commonly used in combination with other types of flour to produce energy-fortified products such as pasta, cereals, bread, crackers, and other bakery products [20]. Pasta is a staple food in Italian cuisine, yet it is a well-known and frequently used product in many countries. Therefore, in various studies, scientists around the world are trying to add amaranth flour or amaranth leaf powder to pasta to increase its nutritional value [21,22]. However, it should be taken into account that adding amaranth flour to pasta increases the nutritional value of the product but, at the same time, changes the technological properties of pasta [23]. The hypothesis of this study is that amaranth as a raw material in the development of new products for vegans can provide the limiting nutrients in the vegan diet such as lysine, methionine, threonine, and iron. This study aimed to evaluate the nutritional value of amaranth pasta and amaranth pasta blends with buckwheat or oat flour and their suitability for a vegan diet by providing the limiting nutrients. The novelty of this study lies in the fact that: (1) organic amaranth grown in Latvia was used in the development of pasta; (2) 100% amaranth pasta was developed as part of the study as a result of the plantain powder that was added for binding purposes as amaranth flour is gluten-free; (3) a detailed analysis of the nutritional value of amaranth pasta and its blends is carried out in accordance with World Health Organization recommendations for healthy nutrition and the regulations of European Parliament and Council, taking into account the nutritional needs of vegans.

## 2. Materials and Methods

Materials: raw organic amaranth seeds (*Amaranthus cruentus* L.) grown in Latvia and produced by individual merchant "Grauds ar dveseli"; raw organic buckwheat seeds (*Fagopyrum esculentum* Moench) grown in Latvia and produced by farm "Bebri"; whole-grain oat (*Avena sativa*) flour (Rigas Dzirnavnieks, Riga, Latvia), beetroot powder (Fity, Latvia), and organic plantain powder (Super Garden, Latvia) were purchased in grocery store. The plantain powder was added for binding purposes as amaranth and buckwheat

flour are gluten-free. The nutritional value and energetic value of materials are reflected in Table 1. The data were taken from the label of the ingredients indicated by the manufacturer.

**Table 1.** Nutritional value and energetic value of ingredients *.

| Nutrients/Energetic Value | Amaranth Seeds | Buckwheat Seeds | Oat Flour | Beetroot Powder | Plantain Powder |
|---|---|---|---|---|---|
| Fat, g | 7.7 | 1.7 | 7.9 | 0 | 0.8 |
| including saturated fatty acids, g | 0.7 | 0.4 | 1.6 | 0 | 0.14 |
| Carbohydrates, g | 61.2 | 70.1 | 55.4 | 57 | 3.7 |
| including sugars, g | 1.3 | 0.3 | 1.2 | 51 | 0.2 |
| Fibre, g | NI | 3.7 | 11.8 | 40 | NI |
| Protein, g | 15.8 | 9.1 | 12.7 | 9 | 2.1 |
| Salt, g | 0.024 | 0.03 | 0.003 | 0.6 | 0 |
| Energetic value, kcal | 377 | 339 | 367 | 265 | 31 |
| Energetic value, kJ | 1594 | 1437 | 1544 | 1110 | 130 |

* Data from food labelling. NI—no information.

Pasta dough preparation for extrusion: amaranth and buckwheat seeds were milled to a fine flour using Mühle 2 (Hawo's, Obrigheim, Germany). According to the recipe (Table 2), beetroot and plantain powders were added to flour and then water was added during further mixing to obtain a dough consistency suitable for extrusion. The amount of water added differed among the samples due to the water absorption capacity of different flours. Buckwheat flour is characterized by a higher water absorption capacity (1.46 g/g) compared to oat (1.26 g/g) and amaranth (0.96 g/g) flour [24], which affected the amount of added water in the samples.

**Table 2.** Recipes of pasta dough and abbreviations of samples, g.

| Ingredients | AMAR | AMAR+BUCKW 15% | AMAR+BUCKW 30% | AMAR+BUCKW 50% | AMAR+OAT 15% | AMAR+OAT 30% | AMAR+OAT 50% |
|---|---|---|---|---|---|---|---|
| Amaranth flour | 150 | 127.5 | 105 | 75 | 127.5 | 105 | 75 |
| Buckwheat flour | - | 22.5 | 45 | 75 | - | - | - |
| Oat flour | - | - | - | - | 22.5 | 45 | 75 |
| Water | 144 | 152 | 167 | 174 | 149 | 158 | 167 |
| Beetroot powder | 10 | 10 | 10 | 10 | 10 | 10 | 10 |
| Plantain powder | 4.5 | 4.5 | 4.5 | 4.5 | 4.5 | 4.5 | 4.5 |

Extrusion: the prepared samples were extruded using the food extruder PCE Extrusiometer L–Serie (Göttfert, Buchen, Germany) where the temperature profile for AMAR was 92/100/106 °C and, for amaranth pasta blends with buckwheat or oat, it was 89/95/102 °C. After extrusion, pasta samples were dried in the convective-rotary oven (SVEBA DAHLAN, Fristad, Sweden) at a temperature of 90 °C for 210 min, followed by cooling to room temperature. As a result, dry pasta samples were obtained, which were used for further analysis (Figure 1). As a control sample, wheat pasta (ingredients: durum wheat semolina, water) was purchased in the grocery.

The research design is given in the Figure 2.

All pasta samples including the control sample were subjected to the following analysis: moisture content was determined using the ISO 712:2009 standard method; total protein content—ISO 1871:2009 standard method; total fibre content—ISO 5498:1981 standard method; fat content—ISO 11085:2009 standard method; fatty acids content—the GC/MS method; starch content—LVS EN ISO 10520:2001 standard method; amino acids content—LVS EN ISO 6498:2012 standard method, iron content—LVS EN 14082:2003 standard method; and sugar content—using the high-performance liquid chromatography (HPLC) method. Carbohydrate content and energetic value were calculated according to EU Regulation No. 1169/2011 of the European Parliament and the Council.

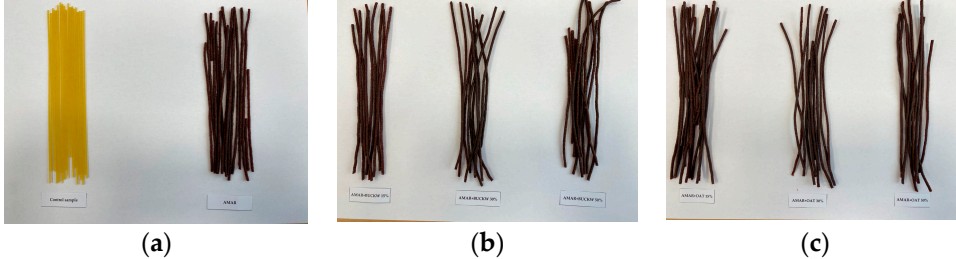

**Figure 1.** Experimental pasta: (**a**) control sample and AMAR; (**b**) amaranth pasta blends with buckwheat flour—AMAR+BUCKW 15%, AMAR+BUCKW 30%, and AMAR+BUCKW 50%; (**c**) amaranth pasta blends with oat flour—AMAR+OAT 15%, AMAR+OAT 30%, and AMAR+OAT 50%.

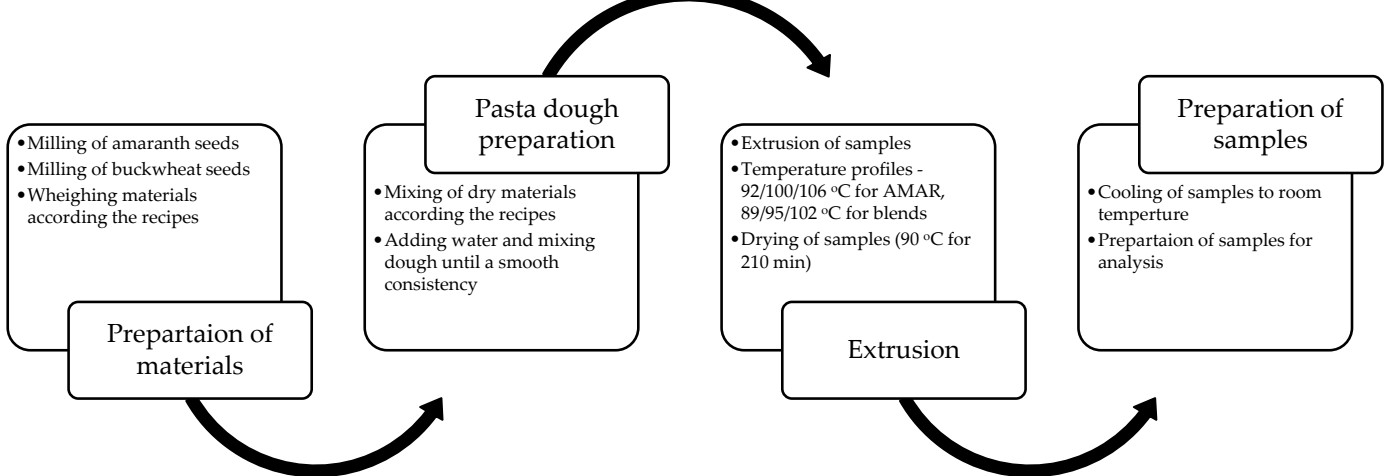

**Figure 2.** The research design.

Statistical analysis: the data reported in the study are average values obtained by repeated measurements for each sample and are expressed as mean value ± standard deviation. The data were processed using statistical methods of Microsoft Office Excel 2013. The significant differences (*p* value < 0.05) of nutrients in the pasta samples were determined with the analysis of variance (ANOVA).

## 3. Results

### 3.1. Nutritional and Energetic Value of Pasta

The nutritional and energetic values of the pasta developed in the study are summarised in Table 3, by regulation No. 1169/2011 on the information's provision to consumers in the labelling of food products [25].

**Table 3.** Nutritional and energetic value per 100 g of amaranth pasta samples developed in the study.

| Nutrients/ Energetic Value | Control Sample | AMAR | AMAR+ BUCKW 15% | AMAR+ BUCKW 30% | AMAR+ BUCKW 50% | AMAR+ OAT 15% | AMAR+ OAT 30% | AMAR+ OAT 50% |
|---|---|---|---|---|---|---|---|---|
| Moisture, g | 11.23 ± 0.17 [a] | 6.75 ± 0.23 [b] | 5.14 ± 0.30 [b] | 5.01 ± 0.27 [b] | 6.33 ± 0.21 [b] | 5.21 ± 0.22 [b] | 5.66 ± 0.23 [b] | 6.42 ± 0.18 [b] |
| Fat, g | 1.04 ± 0.03 [d] | 5.45 ± 0.20 [a] | 4.58 ± 0.24 [b] | 4.37 ± 0.11 [b] | 3.96 ± 0.15 [c] | 4.99 ± 0.14 [b] | 4.77 ± 0.21 [b] | 4.22 ± 0.20 [b,c] |
| including SFA, g | 0.28 ± 0.01 [d] | 1.59 ± 0.02 [a] | 1.13 ± 0.07 [b] | 1.05 ± 0.03 [c] | 1.12 ± 0.06 [b,c] | 1.25 ± 0.03 [b] | 1.16 ± 0.06 [b] | 1.17 ± 0.05 [b] |
| Carbohydrates, g | 69.81 ± 0.90 [a] | 60.77 ± 1.19 [c] | 61.6 ± 1.01 [b,c] | 64.08 ± 1.12 [b] | 63.13 ± 1.45 [b] | 56.76 ± 0.98 [c] | 58.31 ± 1.03 [c] | 59.60 ± 1.09 [c] |
| including sugars, g | 3.95 ± 0.03 [c] | 5.12 ± 0.06 [a] | 4.96 ± 0.14 [a] | 4.88 ± 0.12 [a] | 4.80 ± 0.07 [a] | 5.09 ± 0.16 [a] | 4.30 ± 0.08 [b] | 3.50 ± 0.04 [d] |
| Fibre, g | 0.29 ± 0.01 [e] | 3.49 ± 0.09 [a] | 3.11 ± 0.07 [b] | 2.29 ± 0.09 [c] | 2.99 ± 0.06 [b] | 2.85 ± 0.06 [b] | 2.31 ± 0.07 [c] | 1.91 ± 0.04 [d] |
| Protein, g | 11.78 ± 0.39 [c] | 13.26 ± 0.40 [b] | 13.47 ± 0.34 [b] | 13.47 ± 0.41 [b] | 13.35 ± 0.41 [b] | 13.68 ± 0.35 [b] | 13.87 ± 0.27 [a,b] | 14.35 ± 0.45 [a] |
| Salt, g | 0.007 ± 0.00 [b] | 0.075 ± 0.00 [a] | 0.075 ± 0.00 [a] | 0.063 ± 0.00 [a] | 0.07 ± 0.01 [a] | 0.090 ± 0.00 [a] | 0.078 ± 0.00 [a] | 0.07 ± 0.00 [a] |
| Energetic value, kJ | 1428 ± 21 | 1488 ± 34 | 1471 ± 34 | 1498 ± 30 | 1471 ± 38 | 1405 ± 28 | 1422 ± 28 | 1429 ± 34 |
| Iron, mg | 1.22 ± 0.02 [d] | 6.71 ± 0.11 [a] | 6.10 ± 0.08 [b] | 5.49 ± 0.10 [c] | 4.53 ± 0.18 [c] | 5.68 ± 0.12 [b,c] | 5.90 ± 0.13 [b] | 5.33 ± 0.15 [c] |

Different letters indicate significant differences (*p* < 0.05) within each row.

Significant differences between the control sample and the developed amaranth pasta and its blends with buckwheat or oat flour were identified in all indicators ($p < 0.05$); except for the energy value, there were no significant differences between the samples ($p > 0.05$). The control sample showed a significantly higher carbohydrate content, while the other nutrients had significantly lower values. The developed pasta samples showed similar amounts of moisture and salt, while significant differences were found for carbohydrates, fat, saturated fatty acids, sugars, fiber, and iron. The AMAR pasta sample had a significantly higher amount of fat, saturated fatty acids, fiber, protein, and iron compared to the blends with buckwheat or oat flour. In the AMAR+OAT 50% pasta sample, the significantly highest protein content among the samples was found. The energetic value of all pasta samples ranged from 336 to 354 kcal 100 g$^{-1}$. All seven pasta samples can be evaluated as products with increased nutritional value to which several nutrition claims can be applied by regulation No. 1924/2006 [26]: samples AMAR+BUCKW 15%, AMAR+BUCKW 30%, AMAR+BUCKW 50%, AMAR+OAT 15%, AMAR+OAT 30%, and AMAR+OAT 50% may be labelled low-saturated fat; samples AMAR and AMAR+BUCKW 15% labelled as a source of fibre; and all seven developed samples labelled as the source of protein, low sodium/salt, and high iron. For all samples, the protein content of the total amount of energy was over 15%: AMAR 15.2%, AMAR+BUCKW 15–15.5%, AMAR+BUCKW 30–15.2%, AMAR+BUCKW 50% 15.7%, AMAR+OAT 15–16.5%, AMAR+OAT 30–16.5%, and AMAR+OAT 50–17.1%. The iron content in the samples covered over 30% of the daily reference intake for iron: AMAR 47.9%, AMAR+BUCKW 15–43.6%, AMAR+BUCKW 30–39.2%, AMAR+BUCKW 50–32.4%, AMAR+OAT 15–40.6%, AMAR+OAT 30–42.1%, and AMAR+OAT 50–38.1%. The obtained results can be evaluated as positive because both protein and iron are essential nutrients in a vegan diet. In a cohort study on the daily intake of nutrients for different diet groups, it was found that the protein content of a vegan diet was only 0.99 g/kg of body weight compared to meat eaters' 1.28 g/kg of body weight [3].

### 3.2. Amino Acids, Fats, and Sugar Composition of Pasta

To evaluate the quality of protein in a product, an essential factor is the content of amino acids, since lysine, methionine, and threonine are specified as the limiting amino acids in the vegan diet due to their content in grains [27,28]. The high content of lysine (0.747 g 100 g$^{-1}$) and somewhat high methionine content (0.226 g 100 g$^{-1}$) in amaranth seeds compared to other grains such as wheat, rye, oat, maize, and sorghum contribute to its nutritional quality [18,29]. The amino acid composition of pasta samples can be seen in Table 4.

**Table 4.** Amino acid composition of pasta (g 100 g$^{-1}$).

| Amino Acids | Control Sample | AMAR | AMAR+ BUCKW 15% | AMAR+ BUCKW 30% | AMAR+ BUCKW 50% | AMAR+ OAT 15% | AMAR+ OAT 30% | AMAR+ OAT 50% |
|---|---|---|---|---|---|---|---|---|
| Alanine (Ala) | 0.31 ± 0.01 [a] | 0.46 ± 0.01 [b] | 0.47 ± 0.01 [b] | 0.47 ± 0.01 [b] | 0.49 ± 0.01 [b] | 0.48 ± 0.01 [b] | 0.50 ± 0.00 [b] | 0.53 ± 0.02 [b] |
| Arginine (Arg) | 0.44 ± 0.02 [a] | 1.04 ± 0.02 [b] | 1.06 ± 0.02 [b] | 1.07 ± 0.02 [b] | 1.08 ± 0.03 [b] | 1.06 ± 0.01 [b] | 1.08 ± 0.01 [b] | 1.09 ± 0.02 [b] |
| Asparagine (Asn) | 0.46 ± 0.02 [a] | 0.95 ± 0.02 [b] | 1.02 ± 0.03 [b] | 1.03 ± 0.02 [b] | 1.04 ± 0.03 [b] | 1.02 ± 0.02 [b] | 1.06 ± 0.02 [b,c] | 1.10 ± 0.03 [c] |
| Cysteine (Cys) | 0.24 ± 0.01 [a] | 0.26 ± 0.01 [a] | 0.25 ± 0.00 [a] | 0.25 ± 0.01 [a] | 0.25 ± 0.01 [a] | 0.25 ± 0.00 [a] | 0.26 ± 0.00 [a] | 0.27 ± 0.01 [a] |
| Phenylalanine (Phe) | 0.55 ± 0.02 [a] | 0.48 ± 0.03 [a] | 0.50 ± 0.03 [a] | 0.51 ± 0.02 [a] | 0.52 ± 0.02 [a] | 0.52 ± 0.02 [a] | 0.57 ± 0.0 [a,b] | 0.63 ± 0.04 [b] |
| Glycine (Gly) | 0.37 ± 0.02 [a] | 0.85 ± 0.03 [b] | 0.84 ± 0.02 [b] | 0.83 ± 0.02 [b] | 0.81 ± 0.03 [b] | 0.82 ± 0.02 [b] | 0.80 ± 0.02 [b] | 0.78 ± 0.02 [b] |
| Glutamate (Glu) | 3.75 ± 0.08 [a] | 2.08 ± 0.04 [b] | 2.10 ± 0.03 [b] | 2.15 ± 0.03 [b] | 2.21 ± 0.05 [b] | 2.35 ± 0.05 [c] | 2.45 ± 0.04 [c] | 2.57 ± 0.04 [c] |
| Histidine (His) | 0.22 ± 0.01 [a] | 0.33 ± 0.03 [b] | 0.33 ± 0.02 [b] | 0.33 ± 0.02 [b] | 0.34 ± 0.02 [b] | 0.34 ± 0.02 [b] | 0.35 ± 0.02 [b] | 0.35 ± 0.03 [b] |
| Isoleucine (Ile) | 0.40 ± 0.02 [a] | 0.45 ± 0.02 [a] | 0.46 ± 0.01 [a] | 0.46 ± 0.03 [a] | 0.50 ± 0.03 [a,b] | 0.47 ± 0.02 [a,b] | 0.49 ± 0.02 [a,b] | 0.52 ± 0.02 [b] |
| Leucine (Leu) | 0.78 ± 0.03 [a] | 0.69 ± 0.03 [b] | 0.74 ± 0.02 [a] | 0.75 ± 0.01 [a] | 0.77 ± 0.03 [a] | 0.78 ± 0.02 [a] | 0.83 ± 0.01 [a,c] | 0.88 ± 0.04 [c] |
| Lysine (Lys) | 0.25 ± 0.01 [a] | 0.60 ± 0.03 [b] | 0.60 ± 0.02 [b] | 0.60 ± 0.01 [b] | 0.62 ± 0.03 [b] | 0.60 ± 0.01 [b] | 0.61 ± 0.01 [b] | 0.62 ± 0.02 [b] |
| Methionine (Met) | 0.17 ± 0.01 [a] | 0.26 ± 0.02 [b] | 0.25 ± 0.01 [b] | 0.24 ± 0.01 [b] | 0.24 ± 0.01 [b] | 0.25 ± 0.01 [b] | 0.26 ± 0.01 [b] | 0.26 ± 0.01 [b] |
| Proline (Pro) | 1.29 ± 0.03 [a] | 0.61 ± 0.02 [b] | 0.60 ± 0.02 [b] | 0.61 ± 0.02 [b] | 0.62 ± 0.03 [b] | 0.63 ± 0.02 [b] | 0.67 ± 0.02 [b] | 0.72 ± 0.02 [b] |
| Serine (Ser) | 0.51 ± 0.02 [a] | 0.63 ± 0.03 [b] | 0.63 ± 0.01 [b] | 0.62 ± 0.01 [b] | 0.62 ± 0.01 [b] | 0.63 ± 0.01 [b] | 0.63 ± 0.02 [b] | 0.64 ± 0.02 [b] |
| Tyrosine (Tyr) | 0.25 ± 0.01 [a] | 0.43 ± 0.02 [b] | 0.42 ± 0.01 [b] | 0.42 ± 0.00 [b] | 0.40 ± 0.02 [b] | 0.44 ± 0.01 [b] | 0.48 ± 0.02 [b] | 0.50 ± 0.03 [b] |
| Threonine (Thr) | 0.31 ± 0.01 [a] | 0.43 ± 0.02 [b] | 0.44 ± 0.01 [b] | 0.44 ± 0.22 [b] | 0.47 ± 0.03 [b] | 0.44 ± 0.01 [b] | 0.46 ± 0.02 [b] | 0.50 ± 0.04 [b] |
| Valine (Val) | 0.51 ± 0.03 [a] | 0.54 ± 0.03 [a] | 0.57 ± 0.02 [a] | 0.58 ± 0.01 [a] | 0.63 ± 0.02 [a,b] | 0.60 ± 0.02 [a,b] | 0.64 ± 0.01 [b] | 0.65 ± 0.03 [b] |
| Total essential amino acids | 3.19 | 3.78 | 3.89 | 3.91 | 4.09 | 4.00 | 4.21 | 4.41 |
| Total nonessential amino acids | 7.62 | 7.31 | 7.39 | 7.45 | 7.52 | 7.68 | 7.93 | 8.20 |

Different letters indicate significant differences ($p < 0.05$) within each row.

Comparing the control sample with the developed pasta samples, a significantly higher content of amino acids was found in the developed pasta samples; except for cysteine, phenylalanine, isoleucine, and valine, here, no significant differences were determined. A significantly higher content of glutamate and proline was obtained for the control sample ($p < 0.05$). In addition, it should be emphasized that the content of lysine, methionine, and threonine, which are the limiting amino acids in the vegan diet, was significantly higher in the developed pasta samples compared to the control sample ($p < 0.05$). Significant differences between the amaranth pasta sample and its blends were determined for asparagine, phenylalanine, glutamate, isoleucine, leucine, and valine. The AMAR+OAT 50% sample was characterised by the highest content of total essential amino acids.

In the literature, amaranth seeds are also characterised by their fat profile, indicating that the content of unsaturated fatty acid is 61.0–87.3% of the total fat content [30,31]. It was possible to maintain this positive ratio of fatty acids in the developed pasta (Table 5), where the content of unsaturated fatty acids in sample AMAR was 70.8% of the total fat, in AMAR+BUCKW 15–75.2% of total fat, AMAR+BUCKW 30–75,9%, AMAR+BUCKW 50–71.7%, AMAR+OAT 15–74.9%, AMAR+OAT 30–75.6%, and AMAR+OAT 50–72.3%.

**Table 5.** Fat composition of pasta (g 100 g$^{-1}$).

| Fats | Control Sample | AMAR | AMAR+ BUCKW 15% | AMAR+ BUCKW 30% | AMAR+ BUCKW 50% | AMAR+ OAT 15% | AMAR+ OAT 30% | AMAR+ OAT 50% |
|---|---|---|---|---|---|---|---|---|
| SFA | 0.28 ± 0.01 [d] | 1.59 ± 0.02 [a] | 1.13 ± 0.07 [b] | 1.05 ± 0.03 [c] | 1.12 ± 0.06 [b,c] | 1.25 ± 0.03 [b] | 1.16 ± 0.06 [b] | 1.17 ± 0.05 [b] |
| MUFA | 0.16 ± 0.01 [d] | 1.74 ± 0.08 [a] | 1.45 ± 0.07 [c] | 1.40 ± 0.04 [c] | 1.33 ± 0.04 [c] | 1.59 ± 0.05 [b] | 1.55 ± 0.06 [b] | 1.46 ± 0.07 [c] |
| PUFA | 0.60 ± 0.01 [c] | 2.12 ± 0.09 [a] | 2.00 ± 0.07 [a] | 1.92 ± 0.04 [a] | 1.51 ± 0.04 [b] | 2.15 ± 0.06 [a] | 2.06 ± 0.07 [a] | 1.59 ± 0.07 [b] |
| Total fat | 1.04 ± 0.03 [d] | 5.45 ± 0.20 [a] | 4.58 ± 0.24 [b] | 4.37 ± 0.11 [b] | 3.96 ± 0.15 [c] | 4.99 ± 0.14 [b] | 4.77 ± 0.21 [b] | 4.22 ± 0.20 [b,c] |

Different letters indicate significant differences ($p < 0.05$) within each row.

The developed pasta samples showed a significantly higher fat content compared to the control sample ($p < 0.05$) due to the basic raw material—amaranth flour. In addition, differences in fat content were detected between the blends with buckwheat and oat flour, which was related to the fat content of buckwheat flour (2.31% of the dry matter) and oat flour (5.87% of the dry matter) [24].

Oleic acid (AMAR 28.45%, AMAR+BUCKW 15–31.06%, AMAR+BUCKW 30–31.46%, AMAR+BUCKW 50–31.13%, AMAR+OAT 15–31.47%, AMAR+OAT 30–32.13%, and AMAR+OAT 50–31.88%), linoleic acid (AMAR–28.31%, AMAR+BUCKW 15–27.14%, AMAR+BUCKW 30–27.00%, AMAR+BUCKW 50–26.90%, AMAR+OAT 15–26.56%, AMAR+OAT 30–27.01%, and AMAR+OAT 50–26.62%), and palmitic acid (AMAR–21.79%, AMAR+BUCKW 15–19.74%, AMAR+BUCKW 30–19.64%, AMAR+BUCKW 50–20.64%, AMAR+OAT 15–19.69% AMAR+OAT 30–19.51%, and AMAR+OAT 50–20.85%) were the most abundant amino acids in pasta. The results are related to the findings of a study on amaranth seeds, where linoleic acid (42.00%), oleic acid (30.86%), and palmitic acid (19.88%) were mentioned as the most quantitatively significant fatty acids [31].

No sugar was added to the pasta developed in the study, so Table 6 shows the composition and amount of sugar in the pasta, which is naturally found in the raw materials.

**Table 6.** Sugar composition of pasta (g 100 g$^{-1}$).

| Sugar | Control Sample | AMAR | AMAR+ BUCKW 15% | AMAR+ BUCKW 30% | AMAR+ BUCKW 50% | AMAR+ OAT 15% | AMAR+ OAT 30% | AMAR+ OAT 50% |
|---|---|---|---|---|---|---|---|---|
| Glucose | 0.09 ± 0.00 [b] | 0.14 ± 0.00 [a] | 0.11 ± 0.02 [b] | 0.12 ± 0.01 [b] | 0.10 ± 0.01 [b] | 0.10 ± 0.00 [b] | 0.09 ± 0.00 [b] | 0.08 ± 0.00 [b] |
| Fructose | 0.08 ± 0.00 [b] | 0.15 ± 0.01 [a] | 0.10 ± 0.00 [b] | 0.12 ± 0.00 [b] | 0.13 ± 0.00 [a] | 0.15 ± 0.00 [a] | 0.13 ± 0.00 [a] | 0.09 ± 0.00 [b] |
| Maltose | 2.78 ± 0.00 [a] | 0.06 ± 0.00 [b] | 0.06 ± 0.01 [b] | 0.06 ± 0.00 [b] | 0.04 ± 0.00 [b] | 0.05 ± 0.01 [b] | 0.05 ± 0.00 [b] | 0.05 ± 0.00 [b] |
| Sucrose | 0.42 ± 0.00 [e] | 4.46 ± 0.04 [a] | 4.39 ± 0.10 [a] | 4.30 ± 0.10 [b] | 4.29 ± 0.06 [b] | 4.54 ± 0.13 [a] | 3.81 ± 0.06 [c] | 3.06 ± 0.02 [d] |
| Other sugars | 0.58 ± 0.02 [a] | 0.31 ± 0.00 [b] | 0.30 ± 0.03 [b] | 0.28 ± 0.01 [b] | 0.24 ± 0.00 [c] | 0.25 ± 0.02 [c] | 0.22 ± 0.00 [c] | 0.22 ± 0.01 [c] |
| Total sugar | 3.95 ± 0.03 [c] | 5.12 ± 0.06 [a] | 4.96 ± 0.14 [a] | 4.88 ± 0.12 [a] | 4.80 ± 0.07 [a] | 5.09 ± 0.16 [a] | 4.30 ± 0.08 [b] | 3.50 ± 0.04 [d] |

Different letters indicate significant differences ($p < 0.05$) within each row.

In the control sample, significant differences were found regarding the maltose and sucrose content compared to the other samples ($p < 0.05$). If, in the control sample, 70.4% of

all sugars are provided by maltose, then quantitatively, the majority of the total sugar the in developed samples was sucrose, which accounted for 87.1% of the total sugar in AMAR, 88.5% in AMAR+BUCKW 15%, 88.1% in AMAR+BUCKW 30%, 89.4% in AMAR+BUCKW 50%, 89.2% in AMAR+OAT 15%, 88.6% in AMAR+OAT 30%, and 87.4% in AMAR+OAT 50%. By substituting amaranth flour with oat flour in the ratio of 30% and 50%, it was possible to significantly reduce the total sugar content of the pasta compared to the AMAR sample. The samples—AMAR+BUCKW 15%, AMAR+BUCKW 30%, AMAR+BUCKW 50%, AMAR+OAT 30%, and AMAR+OAT 50%—can be characterised as products with a low sugar content as their content was below 5 g 100 $g^{-1}$ of the product. All samples can be labelled with the nutrition claim of "no added sugar".

## 4. Discussion

The development of new vegan products has become one of the hot topics in the food industry, as the vegan food market reached 23.31 billion US dollars in 2020, and the global market growth of these products was 12.95%, which was the highest increase in recent years [32]. The rapid growth of this market could be related not only to the increase in the number of vegans but also to the desire of consumers to be healthy and to eat healthier by reducing meat consumption and choosing plant-based food products, as a vegan diet can provide health benefits [33].

Pasta made from wheat flour is usually described as a good source of energy, but is poor in protein and lysine content [34], so it is possible to obtain a nutritious product by replacing the basic ingredient or supplementing pasta. The protein content of common pasta from wheat semolina and water (control sample) provided 14.0% of the total amount of energy but with a low content of lysine, methionine, and threonine. The developed amaranth pasta and its blends with buckwheat or oat flour are a good source of protein as they provide more than 15% protein of the total energy with a sufficient content of lysine, methionine, and threonine. The lysine content in the pasta samples was 600–620 mg 100 $g^{-1}$, which should be considered a good source of lysine in the vegan diet because, according to FAO/WHO/UNU recommendations, the lysine requirement is 30 mg/kg of body weight daily [35]. The methionine and threonine content in pasta was 240–260 mg 100 $g^{-1}$ and 430–500 mg 100 $g^{-1}$, respectively, while the requirement of both amino acids is 15 mg/kg of body weight daily [35]. In their study, Aguilar et al. [36] concluded that the new amaranth varieties have amino acid values higher than the requirement established by FAO/WHO/UNU, except for valine, which was determined as a limiting amino acid. The finding for valine cannot be attributed to the pasta obtained in this study, as the valine content was 540–650 mg 100 $g^{-1}$. The daily requirement of valine is 26 mg/kg of body weight [35].

The relatively high amount of protein content in pasta was possible due to the amaranth seeds, whose protein content was 16.8% of the total energy (Table 1), which confirmed, as indicated in literature, that amaranth is a good source of protein and had a higher protein content compared to wheat [22]. This is an important condition, as studies have shown that the protein intake among vegans (including children) is lower compared to meat eaters, covering an average of 13.1% of the total energy [3,37].

Vegetarians, including vegans, are at a higher risk of iron deficiency due to insufficient intake and the low bioavailability from plant products [38]. Therefore, for vegetarians, the recommended intake is 1.8 times higher than for meat eaters, as iron stores in the body are lower or depleted [38,39]. This means that a vegan diet requires products that would be a source of iron. According to regulations No 1924/2006 [26] and No. 1169/2011 [25], a product can be labelled as a source of iron if it contains at least 2.1 mg per 100 g of product. The developed pasta samples contained twice as much, making it possible to label the products with the nutrition claim of "high iron".

One of the healthy eating initiatives is to reduce fat intake and shift from saturated fatty acids to unsaturated fatty acids [40]. Analysing a cohort study's results of a vegan diet, it should be concluded that it is necessary to reduce the total fat, as their content (30.5% of

the total energy) exceeds recommendations [3]. Regarding the content of saturated fatty acids, the vegan diet fulfils the recommendation. The developed pasta samples provided a positive ratio of fatty acids, as the content of unsaturated fatty acids was above 70% of the total fat content. The total fat content of AMAR pasta was over 5 g per 100 g of product due to the fat content in amaranth seeds (7.7 g 100 $g^{-1}$). By substituting amaranth flour with buckwheat flour or oat flour, it was possible to obtain pasta with a significantly lower fat content by maintaining the positive ratio of fatty acids.

The vegan diet is based on the principles of a healthy diet, so the actual issue is the sugar content in the vegan diet and the products used. In a cohort study, analysing the diet of 803 vegans, it was concluded that the content of total sugars was 23.7% of the total energy; compared to meat eaters (23.2% of the total energy), there was no significant difference [3]. The mentioned study did not evaluate the content of free sugars from total sugars; however, the total sugar content of the vegan diet was 113 g per day, which was higher than the reference intake for total sugars of 90 g per day [41]. This means that a vegan diet requires products with a low sugar content, no more than 5 g of sugar per 100 g of product [24], which can be attributed to the AMAR+BUCKW 15%, AMAR+BUCKW 30%, AMAR+BUCKW 50%, AMAR+OAT 30%, and AMAR+OAT 50% pasta. The main source of sugar in the developed pasta was beetroot powder (Table 1), as no sugar was added (Table 2). If there is a need to reduce the sugar content in pasta, the amount of added beetroot powder can be evaluated by reducing it.

## 5. Conclusions

All the developed amaranth pasta samples are suitable for a vegan diet, as they do not contain ingredients or supplements of animal origin, and could be a good addition to a vegan diet, as they provided an increased protein content, low saturated fatty acid content (except AMAR pasta), low sugar content (except AMAR and AMAR+OAT 15% pasta), high iron content, and a sufficient amount of lysine, methionine, and threonine. The hypothesis of this study was confirmed. Nutritious pasta, which satisfies the principles of healthy eating and the needs of a vegan diet, can be obtained by substituting amaranth flour with buckwheat or oat flour. Further research would should a focus on the improvement of amaranth pasta technology to produce instant pasta, thus providing theopportunity to develop disaster food with high nutritional value.

**Author Contributions:** Conceptualisation, I.B.; methodology, I.B.; software, I.B. and A.M.; validation, I.B.; formal analysis, I.B.; investigation, A.M. and I.B.; resources, I.B.; data curation, I.B. and A.M.; writing—original draft preparation, A.M. and I.B.; writing—review and editing, I.B.; visualisation, I.B.; supervision, I.B.; project administration, I.B.; funding acquisition, I.B. All authors have read and agreed to the published version of the manuscript.

**Funding:** This research was funded by the programme "Implementation of the research programme at Latvia University of Life Sciences and Technologies", project No. P21.

**Institutional Review Board Statement:** Not applicable.

**Informed Consent Statement:** Not applicable.

**Data Availability Statement:** Data analyzed during the current study are available from the corresponding author upon reasonable request.

**Conflicts of Interest:** The authors declare no conflict of interest.

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
