# Peer review of "The Potential of Amaranth as a Basic Raw Material for the Production of Pasta for a Vegan Diet"

_applsci, doi:10.3390/app13063944_

Round 1
Reviewer 1 Report
This manuscript decribes the development of pasta with amaranth as an ingredient. However, I am not convinced, because of several aspects/reasons:
1) Although the journal is called Applied Science, I miss the SCIENCE. This comes more like a poduct development study, which is usally rejected by all other scientific journals, as no scientific added value is given...
2) It is stated "Therefore, in various studies, scientists around the world are trying to add amaranth flour or amaranth leaf powder to pasta to increase its nutritional value [19, 20]." So, where is the novelty of the present study ? Or somehow, akind of differentiation to other studies ?
3) Where is a/the scientific hypothesis ?
4) "This study aimed to evaluate the potential of amaranth pasta and amaranth pasta blends with buckwheat or oat flour for a vegan diet to provide the limiting nutrients." ? This aim is too general. What were specific aims with regard to scientific questions in terms of food chemistry, food technology, nutrition or etc. ?
5) To derive a conclusion like "are suitable for a vegan diet" is not scientifcally valid. It was not proven with regard to a basket of goods, an intervention or an observational study. This underlines the more advertisement character of this "study".
All in all, it is more product development than an engineering or natural scientific-driven study.
Author Response
Dear Reviewer,
Thanks for all your comments.
- Although the journal is called Applied Science, I miss the SCIENCE. This comes more like a product development study, which is usually rejected by all other scientific journals, as no scientific added value is given.... All in all, it is more product development than an engineering or natural scientific-driven study.
Author answer: The publication was submitted for publication in the special issue “Bioactive Compounds from Various Sources: Beneficial Effects and Technological Applications II” of the journal Applied Sciences. We do not agree with the reviewer that it is not a science, because according to the tasks of the European Commission - nutrition is one of the 158 scientific activities. Low consumption of fruits, vegetables or fiber, and excess intakes of salt, sugars, and trans and saturated fats are among the top contributors to death and disability caused by non-communicable diseases such as heart disease, diabetes and certain forms of cancer. The increased risk of malnutrition among the European population, which is related to the choice of an unhealthy diet with low protein, vitamin and mineral content, but high carbohydrate and fat content, creates an increasing need for new nutritious products that would be suitable for daily nutrition. It is not engineering but it is an applied science that is essential. In addition, one of the six clusters of the EU Framework Programme "Horizon Europe" (2021 - 2027) is Food, bioeconomy, natural resources, agriculture and environment (6).
- It is stated "Therefore, in various studies, scientists around the world are trying to add amaranth flour or amaranth leaf powder to pasta to increase its nutritional value [19, 20]." So, where is the novelty of the present study? Or somehow, a kind of differentiation to other studies?
Author answer: The novelty of the study: 1) amaranth seeds grown in Latvia were used in the study, as it is known that climatic conditions affect the nutritional value of grains, including pseudo-grains. Latvia is a country in the north of the Europe, therefore the composition of amaranth seeds is different compared to amaranth grown in Asia, Africa or southern Europe, besides the differences in the selected varieties; 2) 100% amaranth pasta was developed as part of the study thanks to the plantain powder that was added for the binding purpose as amaranth flour is gluten-free; 3) A detailed analysis of the nutritional value of amaranth pasta and its blends carried out in accordance with World Health Organization recommendations for healthy nutrition and regulations of European Parliament and Council, taking into account the nutritional needs of vegans.
- Where is a/the scientific hypothesis?
Author answer: The study does not require a scientific hypothesis, considering that this study aimed to evaluate the nutritional value of amaranth pasta and amaranth pasta blends with buckwheat or oat flour and their suitability for a vegan diet providing the limiting nutrients. Hypotheses are more often proposed for master's and doctoral theses.
- "This study aimed to evaluate the potential of amaranth pasta and amaranth pasta blends with buckwheat or oat flour for a vegan diet to provide the limiting nutrients." This aim is too general. What were specific aims with regard to scientific questions in terms of food chemistry, food technology, nutrition or etc.?
Author answer: The aim of the study was specified.
- To derive a conclusion like "are suitable for a vegan diet" is not scientifically valid. It was not proven with regard to a basket of goods, an intervention or an observational study. This underlines the more advertisement character of this "study”.
Author answer: There is no legal definition of ‘vegan’ at EU level when it comes to food. The European Vegetarian Union uses the following definition for vegan food: Vegan foods are not of animal origin and, at no stage of production and processing, has use been made of, or the food been supplemented with, ingredients, processing aids and other substances, whether in processed or unprocessed form, which are of animal origin. Considering that the pasta developed in the study does not contain raw materials of animal origin, and nothing was added in the process of their development, the mentioned products can be considered suitable for vegans. The mentioned sentence has been clarified in the conclusion section of the publication.

Reviewer 2 Report
The manuscript "The potential of amaranth as a basic raw material for the production of pasta for a vegan diet" submitted at Applied Sciences is interesting because the topic of vegan products is a hot topic nowadays. Moreover, the problem with the need for new nutritious products that could cover nutritional deficiencies in the vegan diet is worthy.
In this regard, the manuscript is valuable and is recommendable for publication.
Some minor comments should be taken into account in order to improve the text:
- Please, point out the novelty of the research. The products are new but are there similar on the market or are reported already.
- Table 4 – there is no need of the sign “∑” and the term “Total” at the same time
- - Sensory analysis should be provided and is an important part when the subject is a new food.
- Pictures of the different pasta sample are missing to complete the survey.
- The following references could be useful to expand the introduction and discussion parts - https://jjbs.hu.edu.jo/files/vol13/n3/Paper%20Number%2011.pdf and 10.9755/EJFA.2019.V31.I9.2011.
- How are the % of the different flours chosen?
Author Response
Dear Reviewer,
Thanks for all your comments and suggestions after reviewing our manuscript.
- Please, point out the novelty of the research. The products are new but are there similar on the market or are reported already.
Author answer: The novelty of the research was added in the publication.
- Table 4 – there is no need of the sign “∑”and the term “Total” at the same time.
Author answer: The correction was made in the publication.
- Sensory analysis should be provided and is an important part when the subject is a new food.
Author answer: We agree that sensory evaluation is important, but that was not the purpose of this study. The authors of the study evaluated the obtained pasta as good, however, taking into account that the products are for vegans, the sensory evaluation is planned to be performed among vegans, as it is known that the diet of vegans and their preferences are significantly different from that of omnivores.
- Pictures of the different pasta sample are missing to complete the survey.
Author answer: Authors added the pictures of pastas to the publication.
- The following references could be useful to expand the introduction and discussion parts - https://jjbs.hu.edu.jo/files/vol13/n3/Paper%20Number%2011.pdf and 9755/EJFA.2019.V31.I9.2011.
Author answer: Thank you, the references were added in the publication.
- How are the % of the different flours chosen?
Author answer: The added amount of buckwheat or oat flour in the amaranth pasta blends is listed in Table 2, while the percentage is included in the sample names, for instance, AMAR+BUCKW 15%.

Reviewer 3 Report
Table 1 should be carefully taken, as for nutritional information from labels, there are some calculations and aproximatins performed, there is, this information is not reliable for research purposes. A search for their information from national or international data bases are extremely suggested.
The manuscript possibly was not the final versión, as it had many comments about the format.
The document does not provide a reliable experimental design description for sample or treatment handling.
In table 3. Is energy value giver per 100 g?
Line 154, I recomend the use of one unique unit system for food energy, kJ or Kcal, but do no use both units in the same manuscript.
Although the vegan market is a growing one, it is important to affirm that this new ingredient for pasta can make it suitable for ost people, including vegan and gluten intolerant people.
In general, the disegn of new products or using new raw materials for improving characteristics of the existent ones is a desirable strategy and this paper shows in a proper way the procedure for obtaining this goal. I recomment to improve the description of the statistical analysis, and present the document in the final vesion. Other than this, the paper is suitable for publication
Author Response
Dear Reviewer,
Thanks for all your comments and suggestions after reviewing our manuscript.
- Table 1 should be carefully taken, as for nutritional information from labels, there are some calculations and aproximatins performed, there is, this information is not reliable for research purposes. A search for their information from national or international data bases are extremely suggested.
Author answer: Table 1 has a descriptive function that allows you to get an idea of the nutritional value of the raw materials provided by the manufacturers. In accordance with Regulation (EU) No 1169/2011 of the European Parliament and of the Council of 25 October 2011 on the provision of food information to consumers, the food manufactures are responsible for providing the necessary information, and ensuring it is accurate. The databases have different nutritional values for the same product. Since the raw materials are all grown and produced in Latvia, it was decided that the nutritional value of the raw materials will be taken from the label, as it will be the most accurate.
2. The manuscript possibly was not the final version, as it had many comments about the format.
Author answer: There is some misunderstanding, because the manuscript is prepared in the template provided by the journal.
3. The document does not provide a reliable experimental design description for sample or treatment handling.
Author answer: Authors added the research design in the publication.
- In table 3. Is energy value giver per 100 g?
Author answer: Yes, authors added the information in the publication.
5. Line 154, I recomend the use of one unique unit system for food energy, kJ or Kcal, but do no use both units in the same manuscript.
Author answer: Thank you. The energy value of kcal was deleted.
6. Although the vegan market is a growing one, it is important to affirm that this new ingredient for pasta can make it suitable for most people, including vegan and gluten intolerant people.
Author answer: Since oat flour was also used in the study, it cannot be unequivocally confirmed that it is suitable for people with gluten intolerance.
- In general, the design of new products or using new raw materials for improving characteristics of the existent ones is a desirable strategy and this paper shows in a proper way the procedure for obtaining this goal. I recommend to improve the description of the statistical analysis, and present the document in the final version. Other than this, the paper is suitable for publication.
Author answer: Thank you for comment. The description of the statistical analysis was improved in the publication.

Round 2
Reviewer 1 Report
However, I am not convinced by this revision. Although the opinion of the review was quite drastic, less revision was done in the mansucript. Why were reasons given in the rebuttal not really used to implement into the manuscript and convince finally the readers. It is not about satisfying the reviewer, but improving the manuscript for readers.
In my opinion it is still not scientific enough. Every study has an initial research question - the hypothesis ! Here, it is only a product development study. To only use amaranth from Latvia is not a distinct novelty and justification to expect a new scientific input into the community.
OK. It is a suitable vegan product (absence of animal), but the study is far away from justifying as suitable part of a vegan diet. Here, it becomes obvious that carefulness and awareness of scientific terminology is not quite distinct.
"...this study aimed to evaluate the nutritional value of amaranth..." is not a scientific aim for justifying a publication. This is what each food company does every single day... If at all, this is only of small incremental value for the scientific community....
Usually, incremental work or product development studies are not justified for being published....
Author Response
Dear Reviewer,
Thanks for all your comments.
- However, I am not convinced by this revision. Although the opinion of the review was quite drastic, less revision was done in the manuscript. Why were reasons given in the rebuttal not really used to implement into the manuscript and convince finally the readers. It is not about satisfying the reviewer, but improving the manuscript for readers.
Author answer: We agree that the review was drastic, but the recommendations of review were general, showing the reviewer's opinion about the publication, but not providing constructive and specific recommendations. The publication was improved according to the reviewer's recommendations, adding the novelty of the research to the publication, rewording the aim of the study, providing an explanation in the conclusions about the vegan diet.
2. In my opinion it is still not scientific enough. Every study has an initial research question - the hypothesis! Here, it is only a product development study.
Author answer: Authors added the hypothesis of the study in the publication.
3. To only use amaranth from Latvia is not a distinct novelty and justification to expect a new scientific input into the community.
Author answer: Various studies are being conducted in different countries, where researchers are using raw materials grown in their country, making an assessment of their nutritional value. This allows raw materials of the same species to be compared between countries, which can lead to differences in results, as climatic conditions differ from country to country, which can affect the nutritional value of the product. These studies are essential for the scientific community. They can be the basis for interdisciplinary research.
- It is a suitable vegan product (absence of animal), but the study is far away from justifying as suitable part of a vegan diet. Here, it becomes obvious that carefulness and awareness of scientific terminology is not quite distinct.?
Author answer: The introduction to the study summarizes key findings about the vegan diet, which are based on the results of other studies on limiting nutrients. Next step, the nutritional value of the developed amaranth pasta is evaluated in the context of a vegan diet, evaluating its ability to provide part of the necessary limiting nutrients.
5. ""...this study aimed to evaluate the nutritional value of amaranth..."is not a scientific aim for justifying a publication. This is what each food company does every single day... If at all, this is only of small incremental value for the scientific community.
Author answer: The aim of the research is to focus on the nutritional needs of vegans, evaluating whether the nutritional value of the developed pasta is able to provide the limiting nutrients in the vegan diet. This is an important aspect, taking into account that in healthy diet recommendations, people are invited to reduce meat consumption, to focus more on vegetarian diet. Various European and global initiatives are calling on people to become more sustainable in terms of nutrition as well.
6. Usually, incremental work or product development studies are not justified for being published.
Author answer: We cannot really agree that the research of new products is not binding on the scientific community. Many publications are focused on the development of new products and their evaluation.
